# Visualization of Electrolyte Reaction Field Near the Negative Electrode of a Lead Acid Battery by Means of Amplitude/Frequency Modulation Atomic Force Microscopy

**DOI:** 10.3390/ma16062146

**Published:** 2023-03-07

**Authors:** Yuki Suzuki, Yuki Imamura, Daiki Katsube, Akinori Kogure, Nobumitsu Hirai, Munehiro Kimura

**Affiliations:** 1Graduate School of Engineering, Nagaoka University of Technology, 1603-1 Kamitomioka, Nagaoka 940-2188, Japan; 2Cluster for Pioneering Research, RIKEN, 2-1 Hirosawa, Wako 351-0198, Japan; 3Analytical and Measuring Instruments Division, Shimadzu Corp., 3-25-40 Tonomachi, Kawasaki-ku, Kawasaki 210-0821, Japan; 4Department of Chemistry and Biochemistry, National Institute of Technology (KOSEN), Suzuka College, Shiroko-cho, Suzuka 510-0294, Japan

**Keywords:** additive, atomic force microscopy (AFM), expander, force mapping, interface, lead-acid battery (LAB), lignosulfonate (lignin), negative electrode

## Abstract

The precise observation of a solid–liquid interface by means of frequency modulation atomic force microscopy (FM-AFM) was performed, demonstrating its applicability to a study on lead acid batteries using an electrochemical test cell for in-liquid FM-AFM embedded with a specialized cantilever holder. The consistency and reproducibility of each surface profile observed via amplitude modulation AFM and FM-AFM were verified properly in a strong acidic electrolyte. In terms of FM-AFM, the ability to observe remarkable changes in the force mapping is the most beneficial, especially near the negative electrode surface. The localization of lignosulfonate (LS) added into the electrolyte as an expander could be visualized since this characteristic force mapping was captured when LS was added to electrolyte.

## 1. Introduction

Lead acid batteries (LABs) have been used for more than 150 years [1] and are widely used as invehicle power sources or uninterruptible power supply because of their high thermal reliability, excellent discharge characteristics, and low cost. Such excellent performance based on the stability and reliability of the electrochemical (EC) reaction is the reason for its popularity as a commercial battery product, and the technical details are described in textbooks on LABs. The formation and dissolution of lead sulfate (${\mathrm{PbSO}}_{4}$) crystals in the vicinity of negative electrode during charge–discharge reactions affect the performance of LABs. Here, ${\mathrm{PbSO}}_{4}$ crystals were grown on the negative electrode of LAB during discharge, and ${\mathrm{PbSO}}_{4}$ crystals dissolved during charging, as shown in Equation (Equation 1),
(1)$$\mathrm{Pb}+{\mathrm{SO}}_{4}^{2-}\overset{\mathrm{discharge}}{\underset{\mathrm{charge}}{⇄}}{\mathrm{PbSO}}_{4}+2e^{-}.$$

EC reaction expressed through this simple equation is complex. Lach et al. reported that there are currently over 18,000 studies involving LABs, and the number of published papers is annually growing for each of these subjects [2]. This fact is related to improvements in LABs. Details on improving the electrode structure [3] and expanders such as carbon [4] and lignosulfonate (LS) derivatives [5] have been described. Among many additives, lignin-based materials are widely used as additives for solar cells and various secondary batteries [6,7]. EC characteristics and chemical design guidelines for advanced lignin-based materials have been discussed.

LSs are indispensable additives for the negative electrodes of LABs to achieve excellent performance [8]. LS obtained via the sulfite cooking of lignin, one of the main components of timber, is added in the initial manufacturing step of LABs, and has additive effects such as providing a larger paste viscosity and specific surface area in each process [9]. Various expanders or auxiliary agents have been added to electrodes and/or electrolyte solutions to improve their properties, leading to improvements in the charge/discharge life cycle and a reduction in hydrogen generation as an adverse reaction [10,11,12,13,14]. However, the charge acceptance performance is worsened by the addition of LS. The effect of LSs on negative electrodes has not yet been completely understood [8,9,15], among other improvements. Many researchers are focusing on this clarification as the key to essential performance improvement.

Several evaluation methods for negative electrodes have been demonstrated, such as contact-mode AFM [16,17], scanning electron microscopy (SEM) [18], confocal laser scanning microscopy (CLSM) [19], and transmission X-ray microscopy (TXM) [20,21]. In these studies, when using contact-mode AFM for the in situ observation of the lead electrode surface, the reaction that occurred on the electrode surface could be observed directly [16,17], and SEM images of PbSO${}_{4}$ films on lead showed that the crystallite size was clearly a function of the potential [18]. The CLSM method allows for the examination of large areas (1 mm${}^{2}$ or larger) and can be applied to entire battery plates [19]. The common advantage of the evaluation methods introduced above is their ability to observe the “interface,” which is the boundary between the electrolyte and electrode surface or the PbSO${}_{4}$ crystal surface growing on the electrode surface. If the purpose is to take in situ photographs such as crystal shapes, AFM may not have any advantage over TXM. The common weakness of these methods is that it is impossible to observe the distribution of a reaction field throughout an electrolyte. When the LS was added, the density of PbSO${}_{4}$ crystals formed on the lead electrode after anodic oxidation (discharging) decreased, and the crystals grew [22]. Knehr et al. also suggested a similar tendency for crystal growth by means of TXM [20,21]. However, research on the adsorption site of the expander and the solvation structure in the electrolyte has not significantly progressed, while a number of research papers on morphological changes in the negative electrode during the reaction due to the addition of expander were published [10,11,12,13,14]. Therefore, we conceived the idea that frequency modulation atomic force AFM (FM-AFM) could be used to analyze structural changes at the negative electrode–electrolyte interface [23]. AFM was invented by Binnig et al. in 1986 [24]. AFM offers high resolution, and measurements can be performed in various environments. Subsequently, in 1991, EC-AFM was developed by Manne et al. [25]. AFM measurement modes are divided into contact-mode AFM, which was developed first, and dynamic-mode AFM; which one is used depends on what is being measured. In general, dynamic mode involves a weaker interaction between the probe and the specimen than that of contact mode. Dynamic-mode AFM can be divided into amplitude-modulation AFM (AM-AFM) [26] and FM-AFM. Since dynamic-mode AFM has the advantage of smaller frictional damage to the specimen compared to that of contact-mode AFM, in which the probe touches the specimen, it is widely used in the field of surface science as a nondestructive high-resolution observation method. From the viewpoint of a suitable measurement method for soft samples, it may be necessary to pay attention to peak force tapping (PFT) mode AFM [27], which gives information about biomechanical, biomolecular, and biophysical characteristics on the basis of the force curves and topography. However, it is assumed that PFT-mode AFM is unsuitable for an electrolyte environment with high viscosity due to generally using a cantilever with a low spring constant.

Many interesting processes, such as crystal growth and EC reactions, occur at the solid–liquid interface [28]. For example, a paper by Chen et al. provided details regarding research into metal corrosion using EC-AFM [29]. In 2005, Fukuma et al. successfully performed the atomic-resolution imaging of mica in a liquid environment using FM-AFM [30]. In 2010, the same group developed three-dimensional scanning-force microscopy (3D-SFM or 3D-AFM) and successfully performed the 3D mapping of a saturated solvate of mica [31]. On this basis, there is increasing focus on further applications of surface observation techniques in liquid environments using dynamic-mode AFM (and particularly FM-AFM). However, to extent of our knowledge, there are few examples of such observations with FM-AFM in harsh environments such as in electrolytes that have a low pH and where there are extreme variations in surface shape [29]. The studies that have performed 3D mapping by FM-AFM are even fewer. The reasons for this include the fact that a large amount of damage occurs during the piezo stage and to equipment in a harsh environment, and performing 3D mapping over a wide range is impractical. Furthermore, whereas AM-AFM generates large-amplitude vibrations from the attraction region to the repulsion region, since FM-AFM generates self-excited vibrations with very small amplitudes (compared to AM-AFM) in the attraction or repulsion regions, there is a tendency for the approach to be difficult in cases where there are large structural variations. It is, therefore, thought to be better to perform two-dimensional observations of the XY plane via AM-AFM and the ZX plane via FM-AFM, and then perform three-dimensional analysis of the force field at the solid–liquid interface. Because of this background, it is necessary to perform observations of the same location using both AM and FM modes.

In this study, we fabricated an EC test cell and a specialized cantilever holder that is applicable to FM-AFM for observing the negative electrode–electrolyte interface in an environment that emulates LAB. The sameness and reproducibility of each surface profile observed with AM-AFM and FM-AFM were verified properly in a strong acidic electrolyte. Furthermore, in order to detect the effect of LS on the negative electrode and/or electrolyte, FM-AFM observations on samples added with and without LS at the discharge potential were carried out. As a result, in the case of the LS-added electrolyte, a reaction field that is thought to have been due to the localization of LS was clearly visualized.

## 2. Materials and Methods

Figure 1a shows a schematic of the fabricated EC test cell. In order to visualize the growth and dissolution reactions of PbSO${}_{4}$ crystals that occurred on the negative electrode of LAB, we fabricated theEC test cell for FM-AFM observations. First, a lead (Pb) sheet (purity: 99.999%, thickness: 0.5 mm, Osaka namari-suzu seirensho Co., Ltd., Osaka, Japan) was polished in order to remove discharge and obtain a flat surface. A nonwoven wiper (TechniCloth TX604, Texwipe, Kernersville, NC, USA) coated with a mixture of α-${\mathrm{Al}}_{2}O_{3}$ powder (particle size: 1 µm, Kojundo chemical laboratory Co., Ltd., Sakado, Japan) and silicone oil (Element14 PDMS 200-J, Momentive performance materials Inc., Minato-ku, Japan) was used for polishing. After polishing, the Pb sheet surface was cleaned using the nonwoven wiper coated with 99.5% ethanol, and then cut to give an effective area for the EC reaction of 1 cm${}^{2}$. Next, the Pb sheet was bonded to an acrylic Petri dish. Ultraviolet (UV)-curing adhesive (Photolec A-784-40, Sekisui chemical Co., Ltd., Osaka, Japan) was used as a bonding agent, and was cured via UV illumination with spot UV-curing equipment (Spot-cure SP-9, Ushio Inc., Chiyoda-ku, Japan). Lastly, two Pb wires (purity: 99.99+ %, each length: 8.5 cm (inside: 0.5 cm), PB-241484, Nilaco, Chuo-ku, Japan) were bonded to the Petri dish using the same method.

In order to verify the EC reaction shown in Equation (Equation 1), linear sweep voltammetry (LSV) and cyclic voltammetry (CV) were selected as the EC measurement methods. The Pb sheet being measured was used as the working electrode (WE), and the two Pb wires were used as the reference electrode (RE) and counter electrode (CE). The potential was controlled using a potentiostat (HAB-151A, Hokuto denko Corporation, Meguro-ku, Japan), and the measured values (potential and current) were saved using a multichannel data logger (GL240, Graphtec Corporation, Yokohama, Japan). In the initial stages of all of the experiments, the potential was swept from 0 mV vs. RE to −100 mV vs. RE after dripping the electrolyte, and the initial charging was performed while maintaining the potential for 1 h or more. In the initial charging, the WE formed Pb, and CE formed PbSO${}_{4}$. To identify the shapes in AFM images, AFM observations were performed after sweeping the potential at a constant rate using LSV and stopping the sweep at the desired potential. To determine the effect that adding LS to the electrolyte would have on the electrode, in addition to the same AFM observations, the charge–discharge properties were also analyzed by repeatedly sweeping the potential using CV.

Figure 1b shows a schematic diagram of the LSV/CV measurement setup, and Figure 1c,d show photographs of the customized quartz and cantilever holder (Shimadzu Corporation, Kyoto, Japan). An SPM-8100FM (Shimadzu Corporation, Kyoto, Japan), which can switch between AM and FM modes, was used for the AFM measurements. Only the scanner unit of AFM was installed inside an incubator (CN-40A, Mitsubishi electric engineering Co., Ltd., Chiyoda-ku, Japan) to reduce thermal noise. All experiments were conducted in a cleanroom, and the temperature of the incubator was set to 22 ${}^{\circ }$C. We selected gold-coated cantilever (PPP-NCHAuD, NANOSENSORS, Neuchatel, Switzerland) so that it was resistant to corrosion by dilute sulfuric acid ($H_{2}{\mathrm{SO}}_{4}$). The specific values (in air) of PPP-NCHAuD are as follows: resonance frequency = 330 kHz, force constant = 42 N/m, length = 125 µm, and thickness = 4 µm. In order to reduce the differences between lots as much as possible, a lot of 50 units was purchased, and a new cantilever was used for each experiment. Cantilevers were mounted on corrosion-resistant quartz (customized) that was set into the cantilever holder (customized). A piezo scanner with a large range (X, Y: 30 µm, Z: 5 µm, Shimadzu Corporation, Japan) was selected, so that large crystals could be evaluated. The potential and current density corresponded to the time when the AFM images were saved were obtained from the LSV/CV measurement results that were obtained at the same time.

As the electrolyte, 400 µL of 37% $H_{2}{\mathrm{SO}}_{4}$ (Kishida chemical Co., Ltd., Osaka, Japan) was used. Vanillex-N (Nippon paper industries Co., Ltd., Chiyoda-ku, Japan) was used as a purified, partially desulfonated Na-LS as the additive in the second half of the experiment. This material is used and commercially available as a battery plate expander. The typical chemical properties of Vanillex-N are as follows: LSs = 91%, bulk specific gravity = 0.6, pH (5% solution) = 8.0, ash = 22%, sodium = 10%, and water content = 4%, respectively. We now describe the method for preparing the electrolyte with the addition of Vanillex-N. We first measured out 0.1 g of Vanillex-N using a digital scale (AL204, Mettler toledo, Taito-ku, Japan), and mixed and agitated it with 99.9 mL (g) of purified water. A vortex mixer (rotational speed: 2500 rpm, FLX-F50, AS ONE Corporation, Osaka, Japan) was used for agitation for 30 s or more. Next, 2.0 g of this aqueous solution was measured out using the digital scale, mixed with 98.0 g of 37% $H_{2}{\mathrm{SO}}_{4}$, and agitated as described earlier. Thus, 100 g of Vanillex-N (20 ppm) added solution was prepared as described above. The 400 µL of the prepared solution was used by using a micropipette in all of the experiments involving additives.

## 3. Results and Discussion

The series of experiments were repeated many times to ensure their reproducibility under the same experimental procedure. The following are results of a typical experiment. By demonstrating that the results of a series of CV measurements were qualitatively identical to the results of past studies, we aim to provide evidence that the environment for subsequent AFM experiments was adequate. Figure 2 shows comparisons of the CV measurement results by means of our experimental system with and without the addition of Vanillex-N to the electrolyte. The potential was swept starting from 0 mV vs. RE at a sweep rate of v=3 mV/s in the range of −100 mV vs. RE to +60 mV vs. RE for 200 consecutive cycles. The black dotted line shows the results for the 1st cycle, and the pink solid line shows the results for the 200th cycle. The charge acceptance performance was degraded, and the charge current density peak was reduced by the addition of Vanillex-N, which qualitatively agreed with the previous CV measurement result from Hirai et al. [8,22]. As mentioned in the introduction, this was in exchange with suppressing the adverse reaction. The charge acceptance performance worsened, which is a natural phenomenon when an expander is added.

This suggests that the operation of an actual battery was reproduced, and this experimental setup allowed for us emulate the reactions in an actual negative electrode of LAB.

Figure 3 shows the LSV waveform when the potential was swept to the discharging potential. This experiment was conducted without the addition of Vanillex-N to the electrolyte after initial charging. The potential was swept from the initial charging potential of $E_{0}=-100$ mV vs. RE to the discharging potential of $E_{\mathrm{dis}1}=+30$ mV vs. RE at a sweep rate of v=+0.1 mV/s. After this, the potential was held at $E_{\mathrm{dis}1}$ (for 1 h or more), and AFM observations were performed. The output current density varied from $i_{0}$ to $i_{1}$ due to the discharge reaction, and then decreased to $i_{2}$ when the potential was held at $E_{\mathrm{dis}1}$. We assumed that the reaction converting Pb into PbSO${}_{4}$ had been completed by the time the AFM observations were performed, since $i_{2}=+5$ µA/cm${}^{2}$, and the following AFM observations were carried out. The current density generated when $E_{0}$ was applied is denoted by $i_{0}$; the current density generated when $E_{\mathrm{dis}1}$ was applied is denoted by $i_{1}$; the current density after a time lapse is denoted by $i_{2}$.

Figure 4 shows the AFM results for the negative -electrode–electrolyte interface obtained at the discharging potential. The entire AM-AFM image in the XY plane is seen in Figure 4a (area: 3 µm × 3 µm). Next, line profile analysis was performed (lower parts in Figure 4b–d) in order to obtain the uneven shapes of the upper (Y = 2.5 µm), middle part (Y = 1.5 µm) and lower (Y = 0.5 µm) parts of the XY plane. Immediately after the observations using AM-AFM, we switched to FM-AFM to observe the corresponding ZX plane (area: 800 nm × 3 µm) (inserted figure in Figure 4b–d). In Figure 4, the color shading between the black regions depicted in each inserted figure represents the change in frequency shift, referred to as force mapping. No color shading in the force mapping implies force uniformity in the electrolyte. When the line profiles and ZX planes were compared, the obtained shapes were similar regardless of the observational position. This suggests that the same location could be observed even after switching the operation mode from AM to FM. Furthermore, if we focused on the line profiles on the right-hand side of Figure 4(c1) and the left-hand side of Figure 4(d1), the approaching cantilever tip did not catch up with the surface properly, and an angular granular structure was observed. This is a characteristic shape of PbSO${}_{4}$ reported in previous studies [16,17,20], and there was no significant difference between them. However, in the image where the same location was observed via FM-AFM, the approach was accurate, and the shape was much more curved. As a result, using a combination of AM-AFM and FM-AFM, it was possible to observe fine variations in force mapping that could not be captured by AM-AFM alone as a frequency shift. This allowed for detailed shape analysis to be more simply and accurately performed than that with conventional methods. Here, the equipment was designed so that a large distancing operation between the sample surface and cantilever tip was prevented; only a small distancing operation was performed when switching modes. The series of operations on the build-in software is called “Retract.” Here, we explain the proportional–integral–differential (PID) feedback control of AFM drive system. When approaching under FM mode operation, the I-gain decreased by 200 to 600 compared to the approach under AM mode operation. The I-gain is the optimal value and a peculiar tendency to the instrument. Expecting to properly observe the entire reaction field at the electrode–electrolyte interface with FM-AFM, the scanning range (i.e., Z range) was set to be 1.3 to 1.6 times larger than the topological amplitude of the electrode surface observed with AM-AFM. Whether AM or FM is performed first does not matter, and it was confirmed via multiple experiments that it was possible to observe the same location.

Figure 5 shows the line profile for the force mapping on line PQ extracted from Figure 4(c2).

The profile was asimple rectangle; therefore, it is not possible to determine whether the localized reaction field was too small to be detected or whether the reaction spread throughout the electrolyte homogeneously, as was postulated by Ban et al. [32]. A more detailed discussion, including a comparison with the case of a Vanillex-N added electrolyte, is found below.

Figure 6 shows the LSV waveform when the potential was swept to the discharging potential. A Vanillex-N (20 ppm) solution was used for the electrolyte after initial charging in this experiment. The potential was swept from the initial charging potential of $E_{0}=-100$ mV vs. RE to the discharging potential of $E_{\mathrm{dis}2}=+60$ mV vs. RE at a sweep rate of v=+1 mV/s. After this, the potential was held at $E_{\mathrm{dis}2}$ (for 1 h or more); then, AFM observations were performed. The output current density varied from $i_{3}$ to $i_{4}$ due to the discharge reaction, and then decreased to $i_{5}$ when the potential was held at $E_{\mathrm{dis}2}$. We assumed that the reaction converting Pb into PbSO${}_{4}$ had been completed by the time the AFM observations were performed, since $i_{5}=-1$ µA/cm${}^{2}$; therefore, the following AFM observations were carried out.

Figure 7 shows the AFM images of the negative electrode–electrolyte interface at the discharging potential of the cell using an added Vanillex-N (20 ppm) solution. The effects of Vanillex-N on the negative electrode surface and interface were analyzed using the method proposed above and carried out. When the line profiles and ZX planes (area: 1200 nm × 3 µm) were compared, the same shape was obtained even when the additive was used. However, focusing on the force mapping as shown in Figure 7(d2), a region of abrupt change in color shade that represents the abrupt frequency shift was observed in the immediate vicinity of the negative electrode surface. This result implies a remarkable change in the force mapping near the negative electrode surface. There were two regions of characteristic change observed as a bright color in the immediate vicinity of the electrode surface (denoted by S) and adarker tone on the far side (denoted by R). Two such characteristic regions in force mapping for LAB were first revealed with FM-AFM. This sort of depth profile, which seems to be the distribution of the EC reaction field throughout the electrolyte, could not be found with AM-AFM or TXM. Moreover, we could find the specific regions where EC reactions were not homogeneous, but locally active over the negative electrode. This result provides a clue to the cause of granular structure growth rather than the uniform thickening of the PbSO${}_{4}$ layer.

Figure 8 shows the line profile for the force mapping on line RS extracted from Figure 7(d2). The regions corresponding to the characteristic changes described above are indicated with the red circles in Figure 8. From this figure, a rough change and an abrupt change in the force mapping were recognized as approaching from the electrolyte bulk (R) to the electrode surface (S). This may have been due to the effect of Vanillex-N added to the electrolyte, since this characteristic force mapping was captured only when Vanillex-N was added to the electrolyte. Additionally, such gradation in force mapping did not appear whole over the electrode surface, which implies that Vanillex-N was locally precipitated. During the experiment with the addition of Vanillex-N in the electrolyte, the cantilever tip approach toward the electrode surface was often unstable. This instability may have been due to the fact that the locally precipitated Vanillex-N near the negative electrode surface increased the viscosity of those regions. As a result, the adhesion force became greater than the cantilever restoration force. The exact distribution of Vanillex-N itself was also unclear. In other words, since there was no way to estimate whether Vanillex-N was homogeneously dispersed into the electrolyte during this AFM measurement, it was difficult to confirm whether the segregated Vanillex-N actually precipitated on the negative electrode surface. In summary, the above interpretation consistently explains the series of experimental results.

## 4. Conclusions

AFM observations were performed in a specialized environment that reproduced the electrolyte and negative electrode of LAB, and a comparison with the surface profile obtained via AM-AFM confirmed that it was identical to that obtained by FM-AFM. FM-AFM’s advantage is that it can be used, together with AM-AFM, with existing inliquid observation techniques. The effect of the surface morphology of an electrode with regards to the addition of Vanillex-N, which is commonly used as an expander in actual LAB, was examined.

Using FM-AFM, we could observe the phenomenon in which the added Vanillex-N seemed to be locally precipitated near the electrode surface inhomogeneously. The two characteristic regions observed in force mapping for LAB were first revealed with FM-AFM. It was assumed that a large frequency shift region in the electrolyte near the electrode caused by the addition of Vanillex-N corresponded to a reaction field that helped in improving the characteristics of the negative electrode of LAB. This methodology is expected to play a major role in further research and development and allow for observations under various atmospheres, including in liquids. 

## Figures and Tables

**Figure 1 materials-16-02146-f001:**
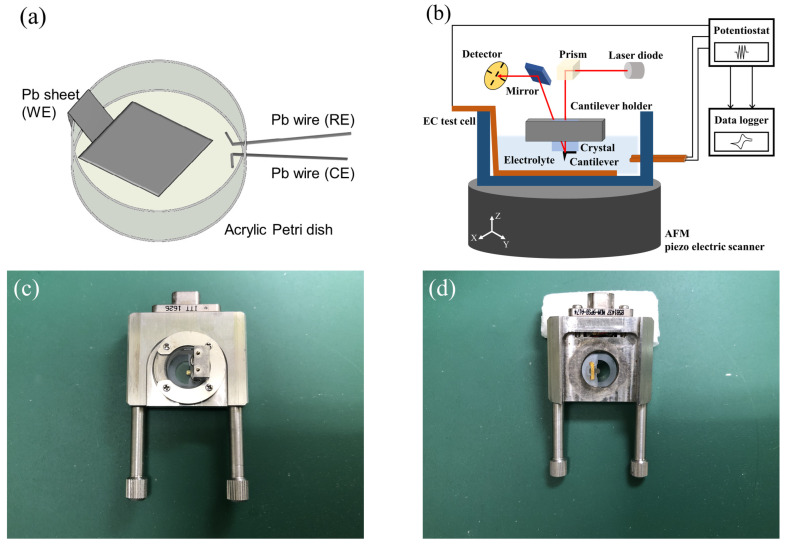
Schematic illustration: (**a**) EC test cell; (**b**) experimental setup for AFM observation with LSV/CV measurement. Photograph of customized quartz and cantilever holder: (**c**) front view; (**d**) rear view.

**Figure 2 materials-16-02146-f002:**
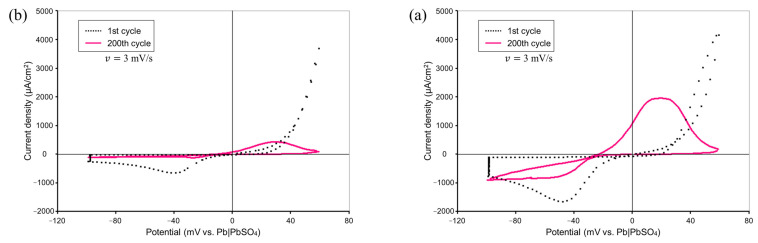
Comparison of CV measurement results (**a**) without and (**b**) with Vanillex-N.

**Figure 3 materials-16-02146-f003:**
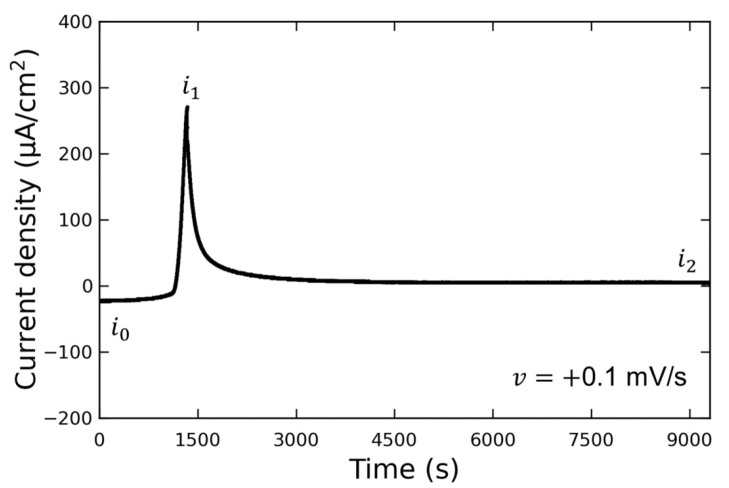
Diagram of the time dependence of current density during discharging potential of lead electrodes in 37% H${}_{2}$SO${}_{4}$.

**Figure 4 materials-16-02146-f004:**
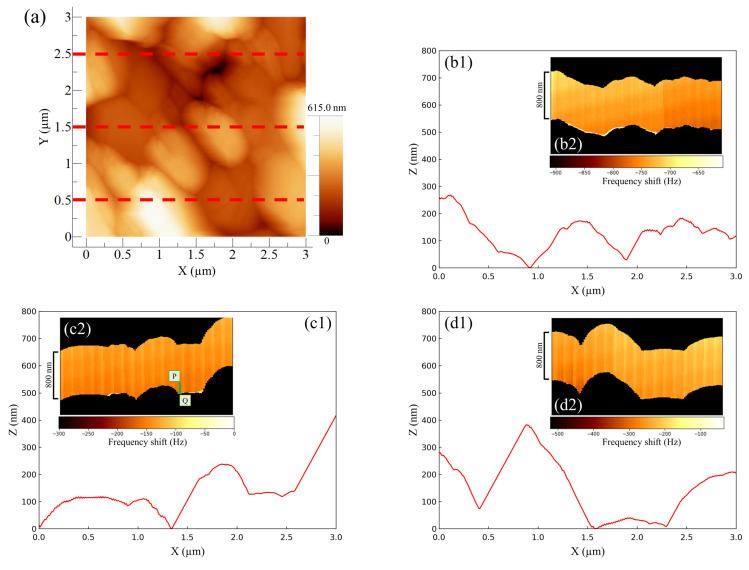
Verifying the identity of AFM image shapes during the discharge potential of lead electrodes in 37% H${}_{2}$SO${}_{4}$: (**a**) XY plane topographic image using AM-AFM; (**b**) offset of Y axis = 2.5 µm; (**c**) 1.5 µm; (**d**) 0.5 µm; (**b1**–**d1**) line profiles analyzed from AM-AFM image; (**b2**–**d2**) ZX plane cross-sectional images using FM-AFM, so-called force mapping, where theresonance frequency and amplitude were 135.9 kHz and 4.8 nm, respectively.

**Figure 5 materials-16-02146-f005:**
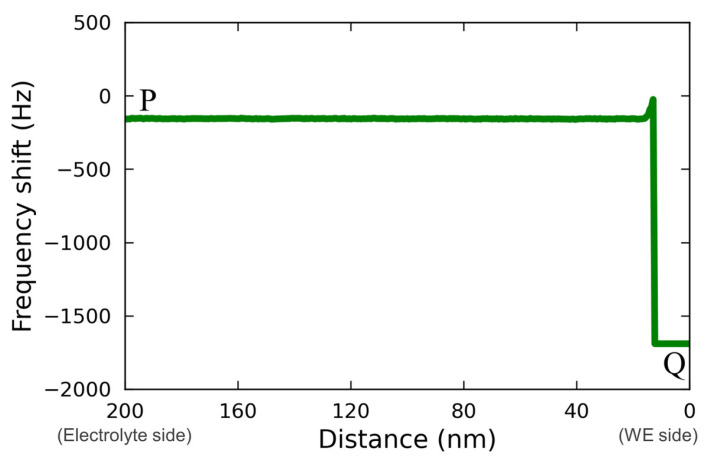
Line profile analyzed from the FM-AFM image of Figure 4(c2).

**Figure 6 materials-16-02146-f006:**
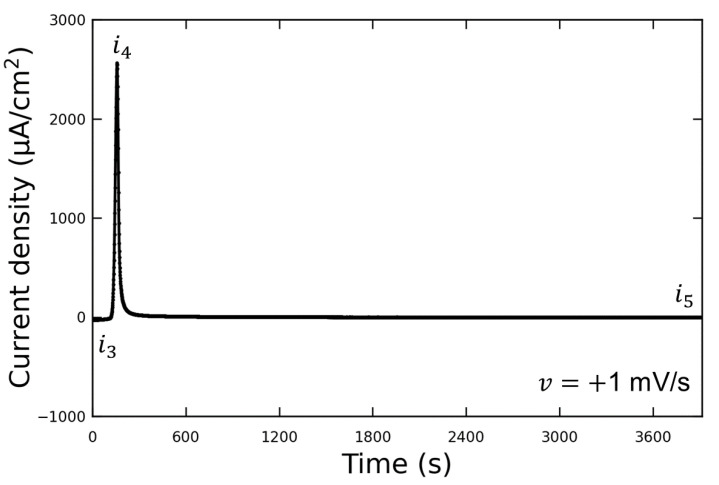
Diagram of the time dependence of current density during discharging potential of lead electrodes in Vanillex-N (20 ppm) added solution.

**Figure 7 materials-16-02146-f007:**
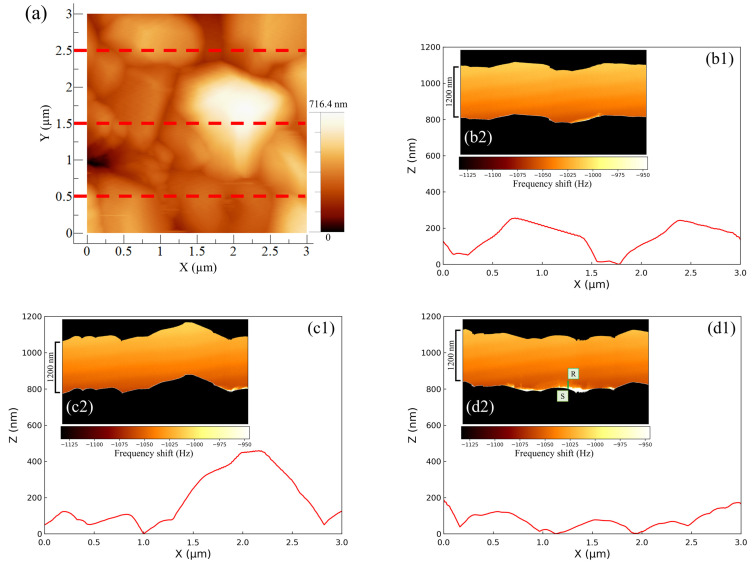
Verifying the identity of AFM image shapes during the discharge potential of lead electrodes in Vanillex-N (20 ppm) added solution: (**a**) XY plane topographic image using AM-AFM; (**b**) offset of Y axis = 2.5 µm; (**c**) 1.5 µm; (**d**) 0.5 µm; (**b1**–**d1**) line profiles analyzed from AM-AFM image; (**b2**–**d2**) ZX plane cross-sectional images using FM-AFM, so-called force mapping, where theresonance frequency and amplitude were 155.8 kHz and 9.3 nm, respectively.

**Figure 8 materials-16-02146-f008:**
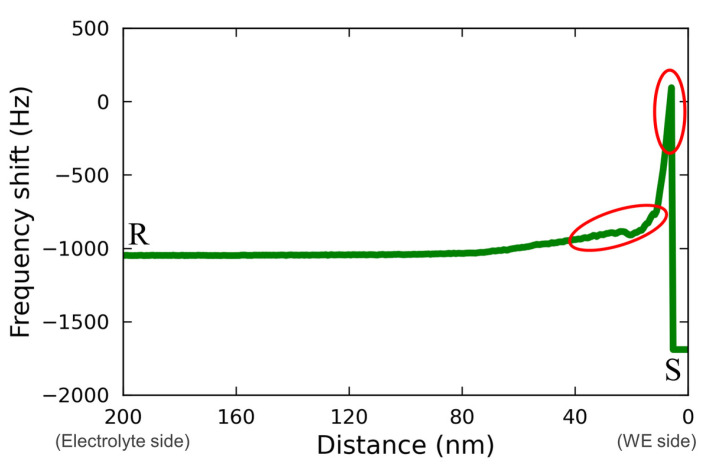
Line profile analyzed from the FM-AFM image of Figure 7(d2).

## Data Availability

Data are contained in the figures of this article. If researchers wish to obtain the original or supporting data, they can contact the author and request the raw data.

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
