# Peer review of "Visualization of Electrolyte Reaction Field Near the Negative Electrode of a Lead Acid Battery by Means of Amplitude/Frequency Modulation Atomic Force Microscopy"

_materials, 2023, doi:10.3390/ma16062146_

Round 1
Reviewer 1 Report
In this work, " Atomic Force Microscopy Imaging of Lignosulfonate Added Electrolyte-Negative Electrode Interface in Lead Acid Battery"
1. The novelty of the presented work should be clearly explained in comparison with published work.
1. Electrochemical data of supercapacitor devices rather than electrodes should be provided.
2. The authors pls carefully calculate the specific capacitance.
3. The stability and reliability of the electrodes and SCs should be reported for evaluation.
Author Response
Detailed responses to the Reviewers
We wish to thank all the reviewers for their kind attention to our manuscript, entitled “Atomic Force
Microscopy Imaging of Lignosulfonate Added Electrolyte-Negative Electrode Interface in Lead Acid
Battery” (by Yuki Suzuki, Yuki Imamura, Daiki Katsube, Akinori Kogure, Nobumitsu Hirai and Munehiro
Kimura), and for their valuable observations. We were pleased to receive positive comments in regard to
our initial submission. Based on the second comment by the Reviewer 4, the title was modified as follows;
“Visualization of Electrolyte Reaction Field Nearby Negative Electrode of Lead Acid Battery by means of
Amplitude/Frequency Modulation Atomic Force Microscopy.” Most of the text in our first manuscript has
been edited in English (see bottom page), but, anyway, once the revised manuscript has been accepted for
publication, we'd like to use the English editing service provided by MDPI. We expect the revised
manuscript will be reviewed in consideration of the fact that it will be edited in English by MDPI after
acceptance. Because, there are no specific indications such as which part of the manuscript is
incomprehensible.
Below we describe the detailed point-by-point revisions in the manuscript in addition to our responses
to the comments by Reviewer 1. Our corrections were added in red.
Yuki Suzuki

Reviewer 2 Report
Very nice design and execution of the work. Good adaptation of the analytical tools to map the interface. There are no technical problems with the paper or the technical content. Good work.
The paper should be reviewed/revised by a native English speaker to enhance the readers understanding of the methodology and results.
Author Response
Detailed responses to the Reviewers
We wish to thank all the reviewers for their kind attention to our manuscript, entitled “Atomic Force Microscopy Imaging of Lignosulfonate Added Electrolyte-Negative Electrode Interface in Lead Acid Battery” (by Yuki Suzuki, Yuki Imamura, Daiki Katsube, Akinori Kogure, Nobumitsu Hirai and Munehiro Kimura), and for their valuable observations. We were pleased to receive positive comments in regard to our initial submission. Based on the second comment by the Reviewer 4, the title was modified as follows; “Visualization of Electrolyte Reaction Field Nearby Negative Electrode of Lead Acid Battery by means of Amplitude/Frequency Modulation Atomic Force Microscopy.” Most of the text in our first manuscript has been edited in English (see bottom page), but, anyway, once the revised manuscript has been accepted for publication, we'd like to use the English editing service provided by MDPI. We expect the revised manuscript will be reviewed in consideration of the fact that it will be edited in English by MDPI after acceptance. Because, there are no specific indications such as which part of the manuscript is incomprehensible.
Yuki Suzuki

Reviewer 3 Report
I can recommend the publication of this manuscript after a minor revision.
1. Minor mistake in affiliation: “....Kawasaki, Kawasaki”...”
2. Write keywords in alphabetical order.
3. The writing must be improved, it is still very poor with numerous wrong terms, typos, or grammar mistakes.
4. Line 27.
You will likely need to re-write your citation sentences, rather than simply replacing the numbers with Authors’ names. This is due to the fact that in order to give readers the maximum appreciation of how your work builds on previous results, each one of the cited sources should be discussed individually and explicitly to demonstrate their significance to your study. We ask that you use the authors' surnames as the subject of a verb, and then state in one or two sentences what they claim, what evidence they provide to support their claim, and how you evaluate their work. We also, therefore, ask that you avoid citing more than one reference in one sentence. This will give you a chance to discuss each reference separately.
What we are asking for is something like this: “Smith (2011) describes the development of a finite element model of hot forging and claims excellent agreement between the model and experiments. However, he tests only one operating condition, tunes his model by modifying the friction coefficient, and compares only the total tool force. A much more detailed comparison would be required to evaluate the precise conditions under which finite element modeling is truly accurate."
5. Line 76: minor mistake – it must be used “...Petri dish....”.
6, Figs. 4 and 7: Can you specify some roughness parameters for this surface microtexture?
7. Explain with more details sentences from lines: 172-174, 177-178, 211-213, 230-231.
8. Lines 243-245: insert university, country.
Authors may consider citing the following reference:
[1] DOI: 10.1002/jemt.22945.
This manuscript can be published after the mentioned revisions.
I can recommend the publication of this manuscript after a minor revision.
Author Response

(The authors gave the same response as above.)

Reviewer 4 Report
Reviewer’s Comments:
The manuscript “Atomic Force Microscopy Imaging of Lignosulfonate Added Electrolyte-Negative Electrode Interface in Lead Acid Battery” is a very interesting work. In this work, Precise observation of solid-liquid interface by means of frequency modulation atomic force microscopy (FM-AFM) was performed and demonstrated its applicability to the study on lead acid battery using electrochemical test cell for in-liquid FM-AFM embedded with specialized cantilever holder. The sameness and reproducibility of each surface profile observed by amplitude modulation AFM and FM-AFM were verified properly in strong acidic electrolyte. It should be noted that what is beneficial with FM-AFM is the ability to observe a remarkable change in the force mapping especially nearby the negative electrode surface. While I believe this topic is of great interest to our readers, I think it needs major revision before it is ready for publication. So, I recommend this manuscript for publication with major revisions.
1. In this manuscript, the authors did not explain the importance of the Interface in the introduction part. The authors should explain the importance of Interface.
2) Title: The title of the manuscript is not impressive. It should be modified or rewritten it.
3) Correct the following statement “It seems to be the first time that the localization of lignosulfonate (LS) added in electrolyte as an expander could be visualized since this characteristic force mapping was captured when LS was added in electrolyte”.
4) Keywords: The Interface is missing in the keywords. So, modify the keywords.
5) Introduction part is not impressive. The references cited are very old. So, Improve it with some latest literature like 10.3390/molecules27217368, 10.3390/pr10081455
6) The authors should explain the following statement with recent references, “It is thought 132
that the charge acceptance performance was degraded and the charge current density peak 133
was reduced by the addition of Vanillex-N”.
7) Add space between magnitude and unit. For example, in synthesis “21.96g” should be 21.96 g. Make the corrections throughout the manuscript regarding values and units.
8) The author should provide reason about this statement “In addition, in the FM approach, the current gain was set to around 200 to 600 times smaller than that for the AM approach, and the scanning range was set to 1.3 to 1.6 times larger”.
9. Comparison of the present results with other similar findings in the literature should be discussed in more detail. This is necessary in order to place this work together with other work in the field and to give more credibility to the present results.
10) Conclusion part is very long. Make it brief and improve by adding the results of your studies.
11) There are many grammatic mistakes. Improve the English grammar of the manuscript.
Author Response
Detailed responses to the Reviewers
We wish to thank all the reviewers for their kind attention to our manuscript, entitled “Atomic Force Microscopy Imaging of Lignosulfonate Added Electrolyte-Negative Electrode Interface in Lead Acid Battery” (by Yuki Suzuki, Yuki Imamura, Daiki Katsube, Akinori Kogure, Nobumitsu Hirai and Munehiro Kimura), and for their valuable observations. We were pleased to receive positive comments in regard to our initial submission. Based on the second comment by the Reviewer 4, the title was modified as follows; “Visualization of Electrolyte Reaction Field Nearby Negative Electrode of Lead Acid Battery by means of Amplitude/Frequency Modulation Atomic Force Microscopy.” Most of the text in our first manuscript has been edited in English (see bottom page), but, anyway, once the revised manuscript has been accepted for publication, we'd like to use the English editing service provided by MDPI. We expect the revised manuscript will be reviewed in consideration of the fact that it will be edited in English by MDPI after acceptance. Because, there are no specific indications such as which part of the manuscript is incomprehensible.
Below we describe the detailed point-by-point revisions in the manuscript in addition to our responses to the comments by Reviewer 4. Our corrections were added in red.
Yuki Suzuki

Reviewer 5 Report
Thanks for the chance to assess the paper. I believe that the paper cannot be considered satisfactory for the journal based on a number of factors.
* The study demonstrates very limited novelties . The authors need to comprehensively summarize the progress in this field.
* The paper just present data, where there is very little discussion in the paper: more must be given
* There are problems with the standard of English. This MUST be corrected if the paper is to be published.
* The Introduction is short, weak, and uncritical. The state of the art is not covered, and hence, the impact and relevance of the present work is not in any context.
* Insufficient experimental information given
Author Response
Detailed responses to the Reviewers
We wish to thank all the reviewers for their kind attention to our manuscript, entitled “Atomic Force Microscopy Imaging of Lignosulfonate Added Electrolyte-Negative Electrode Interface in Lead Acid Battery” (by Yuki Suzuki, Yuki Imamura, Daiki Katsube, Akinori Kogure, Nobumitsu Hirai and Munehiro Kimura), and for their valuable observations. Based on the second comment by the Reviewer 4, the title was modified as follows; “Visualization of Electrolyte Reaction Field Nearby Negative Electrode of Lead Acid Battery by means of Amplitude/Frequency Modulation Atomic Force Microscopy.” Most of the text in our first manuscript has been edited in English (see bottom page), but, anyway, once the revised manuscript has been accepted for publication, we'd like to use the English editing service provided by MDPI. We expect the revised manuscript will be reviewed in consideration of the fact that it will be edited in English by MDPI after acceptance. Because, there are no specific indications such as which part of the manuscript is incomprehensible.
Below we describe the detailed point-by-point revisions in the manuscript in addition to our responses to the comments by Reviewer 4 and Reviewer 5. Our corrections were added in red.
Yuki Suzuki

Round 2
Reviewer 5 Report
· The study demonstrates very limited novelties .
· A number of investigators studied the effect of lignosulfonates in negative electrode active mass NAM in recent decades such as DOI 10.1149/1.2127655 and doi:10.1016/j.jpowsour.2008.10.09
* The paper just present data, where there is very little discussion in the paper:
** The Introduction is short, weak, and uncritical. The state of the art is not covered, and hence, the impact and relevance of the present work is not in any context.
· Insufficient experimental information given such as electrochemical test
· Not all the comments were addresses by authors
Author Response
We sincerely thank you for taking bunch of time for reviewing our manuscript.
We believe that the reviewer would understand the major objective in our paper is to propose the outstanding feature of FM-AFM rather than the discussion on the nature of lignosulfonate.
The novelty of this paper is that the depth profile of the electrochemical reaction field can be estimated by FM-AFM, which was described in Line 75-78 and Line 236-253. For the purpose of making clear our objective easier to understand, the comparison without or with the addition of lignosulfonate is made. It would be unfair to be judged unworthy of publication merely because the lignosulfonate innovation was not detailed. Indeed, two articles;
DOI 10.1149/1.2127655 and doi:10.1016/j.jpowsour.2008.10.09,
were surveyed. Former article was found, and is added as a reference following the recommendation of editor. However, the latter was not found probably because the DOI is not correct. As for “the effect of lignosulfonates in negative electrode active mass NAM,” we already cited appropriate article as reference #13 in our original manuscript. In any case, the authors don't think the quotes of lignosulfonate are inadequate.
The following critiques are unchanged from Round 1;
The paper just present data, where there is very little discussion in the paper:
** The Introduction is short, weak, and uncritical.
The state of the art is not covered, and hence, the impact and relevance of the present work is not in any context.
Insufficient experimental information given such as electrochemical test
The reviewer5 thinks that "not all the comments were addresses by authors" whereas other reviewers have confirmed the revision.
The author wishes that the reviewer would notice the manuscript was revised properly when the reviewer read carefully the reply letter for round1 as well as the revised manuscript.
Best regards,
Yuki Suzuki